

# Evaluation of snow processes over the Western United States in E3SM land model

Dalei Hao[1], Gautam Bisht[1], Karl Rittger[2], Timbo Stillinger[3], Edward Bair[3], Yu Gu[4], L. Ruby Leung[1]

[1]Atmospheric Sciences and Global Change Division, Pacific Northwest National Laboratory, Richland, WA, USA Laboratory
[2]Institute for Arctic and Alpine Research, University of Colorado, Boulder, CO, USA
[3]Earth Research Institute, University of California, Santa Barbara, CA, USA
[4]Joint Institute for Regional Earth System Science and Engineering and Department of Atmospheric and Oceanic Sciences, University of California, Los Angeles, CA, USA

*Correspondence to*: dalei.hao@pnnl.gov

**Abstract.** Seasonal snow has crucial impacts on climate, ecosystems and humans, but it is vulnerable to global warming. The land component (ELM) of the Energy Exascale Earth System Model (E3SM), mechanistically simulates snow processes from accumulation, canopy interception, compaction, snow aging to melt. Although high-quality field measurements, remote sensing snow products and data assimilation products with high spatio-temporal resolution are available, there has been no systematic evaluation of the snow properties and phenology in ELM. This study comprehensively evaluates ELM snow

simulations over the western United States at 0.125° resolution during 2001-2019 using the Snow Telemetry (SNOTEL) in situ networks, MODIS remote sensing products (i.e., MCD43 surface albedo product, the spatially and temporally complete (STC) Snow-Covered Area and Grain Size (MODSCAG) and MODIS Dust and Radiative Forcing in Snow (MODDRFS) products (STC-MODSCAG/STC-MODDRFS), and the Snow Property Inversion from Remote Sensing (SPIReS) product) and two data assimilation products of snow water equivalent and snow depth (i.e., University of Arizona (UA) and SNOw

Data Assimilation System (SNODAS)). Overall the ELM simulations are consistent with the benchmarking datasets and reproduce the spatio-temporal patterns, interannual variability and elevation gradients for different snow properties including snow cover fraction ($f_{sno}$), surface albedo ($\alpha_{sur}$) over snow cover regions, snow water equivalent (SWE) and snow depth ($D_{sno}$). However, there are large biases of $f_{sno}$ with dense forest cover and $\alpha_{sur}$ in the Rocky Mountains and Sierra Nevada in winter, compared to the MODIS products. There are large discrepancies of snow albedo, snow grain size and light-absorbing particles

induced snow albedo reduction between ELM and the MODIS products, attributed to uncertainties in the aerosol forcing data, snow aging processes in ELM, and remote sensing retrievals. Against UA and SNODAS, ELM has a mean bias of -20.7mm (-35.9%) and -20.4 mm (-35.5%), respectively for spring, and -13.8 mm (-27.8%) and -10.2 mm (-22.2%), respectively for winter. ELM shows a relatively high correlation with SNOTEL SWE, with mean correlation coefficients of 0.69, but negative mean biases of -122.7 mm, respectively. Compared to the snow phenology of STC-MODSCAG and SPIReS, ELM shows

delayed snow accumulation onset date by 17.3 and 12.4 days, earlier snow end date by 35.5 and 26.8 days, and shorter snow duration by 52.9 and 39.5 days. This study underscores the need for diagnosing model biases and improving ELM





representations of snow properties and snow phenology in mountainous areas for more credible simulation and future projection of mountain snowpack.

## 1 Introduction

Snow, a key component of the cryosphere, has a large influence on the terrestrial energy budget and water and carbon cycles (Berghuijs et al., 2014; Niittynen et al., 2018). With high albedo and low thermal conductivity, snow also affects regional climate (Flanner et al., 2011; Henderson et al., 2018; Skiles et al., 2018). Under global warming, less precipitation will fall as snow and snow will melt earlier (Barnett et al., 2005), which will have large impacts on water availability in snow-dominated regions (Barnett et al., 2005; Musselman et al., 2021). Climate models project the snow water equivalent (SWE) declines of

~25% by 2050 for the Western United States (WUS; see Table A1 for acronyms and symbols used in the study) (Musselman et al., 2021; Siirila-Woodburn et al., 2021), with large impacts on ecosystem function, wildlife habitats, flood hazard, tourism, recreation and socio-economic activities (Hamlet and Lettenmaier, 2007; Mameno et al., 2022). Accurately characterizing and projecting future changes in snow processes and timing of these changes is crucial for planning our response to climate change.

Numerous parameterizations and models with various degrees of complexity have been developed to simulate seasonal snow dynamics and improve our understanding of snow processes (Krinner et al., 2018; Lee et al., 2021; Magnusson et al., 2015). These parameterizations/models have been coupled to land surface models (LSMs) (Krinner et al., 2018) to represent snow grain particles (Räisänen et al., 2017), snow cover (Swenson and Lawrence, 2012), snow albedo (Flanner et al., 2007), snowpack compaction (Decharme et al., 2016), and snow interception by vegetation (Lundquist et al., 2021). The Energy

Exascale Earth System Model (E3SM) Land Model (ELM) (Leung et al., 2020) includes a multi-layer snow scheme to simulate the prognostic snow processes such as snow accumulation, snow interception, snow compaction, and snow melt. Recently, the snow albedo model in ELM was improved to include new radiative transfer solvers with improved accuracy (Dang et al., 2019), add non-spherical snow grain shape (Hao et al., 2022a), account for the internal mixing of light-absorbing particles (LAPs) with snow (Böttcher et al., 2014); Hao et al., 2022a), and incorporate new parameterizations to account for the sub-

grid topographic effects on solar radiation (Hao et al., 2021; Hao et al., 2022b) (see Section 2.1 for details). With these enhancements and improvements, ELM may skillfully simulate snow dynamics at a regional scale (e.g., WUS).

Previous studies evaluated simulations of snow cover fraction ($f_{sno}$), SWE, snow depth ($D_{sno}$) (Toure et al., 2018; Toure et al., 2016) and snowmelt timing (Toure et al., 2018) in the Community Land Model V4 (CLM4) in the Northern Hemisphere at a

coarse spatial resolution of 0.5°×0.67°. The 0.25° simulations of surface albedo ($\alpha_{sur}$), $f_{sno}$ and SWE in the Canadian Land Surface Scheme (CLASS) were evaluated over eastern Canada (Verseghy et al., 2017), but snow phenology was not assessed. Monthly SWE in the 1° coupled land-atmosphere simulations of E3SM v1 was evaluated over the Contiguous United States by (Brunke et al., 2021), who attributed SWE uncertainties to the biases in temperature and precipitation. Overall, previous



studies only evaluated a few snow variables in LSMs mostly at coarse spatial resolutions (Table A2), although more high-
resolution remote sensing observations and data assimilation products of snow variables (e.g., snow albedo ($\alpha_{sno}$), snow grain
size ($S_{sno}$) and snow albedo reduction induced by LAPs in snow ($R_{sno}$)) have become available. The snow phenology in LSMs
has rarely been evaluated explicitly and how LSMs capture the interannual variability of snow variables and how those
variables vary along an elevation gradient have not been well investigated.

A series of high-quality field snow measurements, remote sensing and data assimilation snow datasets/products with high
spatio-temporal resolution are available over the WUS. The *in situ* Snow Telemetry (SNOTEL) stations widely distributed
across the WUS provide long-term SWE field measurements (Serreze et al., 1999). Optical remote sensing data has been
widely used to map snow dynamics (Dietz et al., 2012; Dong, 2018). The Moderate Resolution Imaging Spectroradiometer
(MODIS) reflectance data at 463 m spatial resolution have been used to retrieve multiple key snow-related variables
including $\alpha_{sur}$ (Schaaf et al., 2002), $f_{sno}$ (Bair et al., 2021c; Painter et al., 2009), $\alpha_{sno}$, $S_{sno}$, and $R_{sno}$ (Bair et al., 2021c; Painter
et al., 2012). These MODIS data accurately capture snow dynamics during accumulation and melt (Rittger et al., 2013;
Wang et al., 2018) and the high daily temporal resolution of these datasets is helpful for capturing rapid snow variations.
Some available remote sensing snow phenology products (Chen et al., 2015; Metsämäki et al., 2018; Takala et al., 2009)
adopt different optical or microwave satellite observations to extract snow phenology date and duration. Besides, they use
different snow phenology definitions, and include different snow phenology metrics, which can affect their use as a
reference. Alternatively, the same phenology extraction methods can be used to derive snow phenology metrics for both
LSMs and MODIS daily $f_{sno}$ data, avoiding inconsistencies of definitions and extraction methods. Data assimilated SWE and
snow depth ($D_{sno}$) products are also available that integrate field measurements, remote sensing observations, and model
simulations (Center, 2004; Zeng et al., 2018). These data assimilation products have high spatial resolution of <5 km and
higher reliability over mountainous and forested regions due to the constraints of in situ networks (Dawson et al., 2018).
These datasets provide good opportunity for comprehensively evaluating the accuracy of snow variables and snow
phenology in LSMs.

The aim of this study is to systematically evaluate the high-resolution 0.125° ELM simulations of key snow variables and
snow phenology over the WUS, using in situ, remote sensing and data assimilation snow products. Specifically, offline ELM
simulations with new improvements related to snow processes over the WUS were conducted during 2001-2019. Field snow
measurements, three MODIS remote sensing products, and two data assimilation snow products were collected as
benchmarking datasets for the ELM simulations (see Section 2.3 for details). All the ELM outputs and benchmarking datasets
were regridded to an identical spatio-temporal resolution of 0.125° and daily. Snow properties variables including $\alpha_{sur}$, $f_{sno}$,
$\alpha_{sno}$, $S_{sno}$, $R_{sno}$, SWE and $D_{sno}$ were used in the analysis. Multiple snow phenology metrics were derived from both ELM and
remote sensing products using the same definitions and extraction methods (see Section 2.4 for details). The spatial patterns,
temporal correlations, interannual variabilities, elevation gradients, and change with forest cover of snow properties and snow



phenology in ELM were evaluated against the benchmarking datasets. Uncertainties in the ELM and benchmarking datasets, implications for model improvements and limitations of the study are discussed.

## 2 Materials and Methods

### 2.1 Model description

ELM, the land component of E3SM, originates from the Community Land Model Version 4.5 (CLM4.5) (Golaz et al., 2019). ELM uses a multi-layer scheme (up to 5 layers by default) to dynamically simulate various snow processes, e.g., snow accumulation, melting, aging (i.e., the evolution of snow grain size), compaction, metamorphism, aerosol deposition and redistribution, and canopy snow interception and unloading. Specifically, ELM uses the Snow, Ice, and Aerosol Radiative (SNICAR) model to calculate snow albedo and vertically-resolved absorption of solar radiation, considering the evolving snow grain size, solar zenith angles (SZAs), sky conditions, underlying background and snow impurities (e.g., black carbon (BC) and dust) (Flanner et al., 2007). ELM uses the snow water equivalent (SWE) and standard deviation of elevation to estimate snow cover fraction ($f_{sno}$). The hysteresis of snow accumulation and ablation is also accounted for in ELM (Swenson and Lawrence, 2012).

Compared to CLM4.5, some key updates related to snow processes have been included in ELM. First, the original SNICAR model has been replaced by a hybrid model (SNICAR-AD) of SNICAR and delta-Eddington adding–doubling radiative transfer solver, which corrects the snow albedo bias for large SZAs and can better represent the shortwave radiative properties of snow (Dang et al., 2019). Second, compared to only external mixing in CLM4.5, both external mixing and internal mixing of hydrophilic BC-snow and dust-snow are now represented in ELM (Hao et al., 2022a; Wang et al., 2020). Third, the direct and diffuse irradiance under different atmospheric profiles and their dependence on SZA are included (Hao et al., 2022a). Fourth, the effects of non-spherical snow grain shape on snow albedo are considered (Hao et al., 2022a). Fifth, a new parameterization of sub-grid topographic effects on solar radiation has been implemented in ELM to account for the impacts of macro-scale shadow, occlusion and multi-scattering between adjacent terrain on surface albedo (Hao et al., 2021; Hao et al., 2022b).

### 2.2 Model setup and experiment design

Selected for this study, the WUS has heterogeneous topography with diverse elevations ranging from 0 to above 3 km (Figure 1a). The WUS includes three major mountain ranges: the Cascades Range, Sierra Nevada, and Rocky Mountains, which are characterized by frequent snow cover. The elevation data was acquired from the Shuttle Radar Topography Mission (SRTM) DEM dataset (Rabus et al., 2003). The forest cover data in 2010 shown in Figure 1b was acquired from the 30 m Landsat Vegetation Continuous Fields (VCF) tree cover datasets derived from the GFCC Surface Reflectance product (Sexton et al., 2013). Both the DEM and forest cover data were aggregated to 0.125° using the area-weighted average method. For analysis,



elevations were divided into different intervals (see Figure 1c). Elevations less than 0.5 km are not included in the statistical
analysis as snow cover is close to 0. The forest cover was divided into five levels (see Figure 1d). The area fractions of different
intervals of elevation and forest cover are shown in Figure 1c and 1d, respectively.

ELM simulations at 0.125° spatial resolution were conducted over the WUS from 1979 to 2019 driven by hourly
meteorological forcing data from the National Land Data Assimilation System phase 2 (NLDAS-2) with spatial resolution of
0.125° (Xia et al., 2012). Specifically, the prescribed satellite phenology (SP) mode was used with input of MODIS leaf area
index data (Myneni et al., 2002). The climatological monthly aerosol deposition data (e.g., black carbon and dust) with a spatial
resolution of 1.9°×2.5° from the Community Atmosphere Model version 5 coupled with chemistry (Lamarque et al., 2010)
was used, which was temporally and spatially downscaled to half-hourly and 0.125° using bilinear interpolation. For the snow
albedo module, SNICAR-AD was configured with: 1) the SZA-dependence solar irradiance under the mid-latitude winter
atmosphere, 2) spherical snow grain shape, 3) internal mixing of hydrophilic BC-snow, (4) external mixing of dust-snow, and
(5) neglect of organic carbon due to its high uncertainties. The sub-grid topographic effects on solar radiation were included
in the ELM configuration. The model was run at a half-hourly step. The first 31-year run from 1979 to 2000 was used to spin-
up the model to reach equilibrium and then the remaining 19-year run (i.e. 2001-2019) was used in the analysis. The variables
of interest were output at half-hourly, daily and monthly scales.


**Figure 1: Spatial distributions of (a) elevation and SNOTEL sites (grey points) and (b) forest cover over the WUS, and the area proportions of different (c) elevation and (d) forest cover intervals. The Cascades Range, Sierra Nevada, and Rocky Mountains are highlighted in panel (b).**

**2.3 Benchmarking datasets**

*In situ* Bias Correction and Quality Control (BCQC) SNOTEL daily SWE data from 2001-2019 (Table 1) were used as the benchmarking dataset to evaluate the performance of ELM. SNOTEL stations, operated by the U.S. Department of Agriculture Natural Resources Conservation Service (NRCS) provide long-term, widely-distributed and high-quality field measurements of SWE across the WUS (https://www.nrcs.usda.gov/). BCQC SNOTEL eliminated data outliers and erroneous values, fixed

the inconsistencies of different variables, and corrected the bias of the raw data (Sun et al., 2019; Yan et al., 2018). Specifically, 788 SNOTEL sites in the WUS were included in the study (Figure 1a).



Three daily 463 m MODIS-based remote sensing products from 2001-2019 were used to evaluate the performance of ELM (Table 1). The first one is the MCD43A3 surface albedo version 6 product (named as MCD43 hereafter). The MCD43 product
provides black-sky and white-sky surface albedo at local solar noon (Schaaf et al., 2002), which could well capture the snow effects on $\alpha_{\mathrm{sur}}$ (Wang et al., 2018). This dataset represents the albedo of the entire MODIS pixel which could include vegetation or soil if the observed pixel is not 100% snow cover, and thus it will underestimate snow albedo for fractionally covered pixels as vegetation and soil have darker broadband albedos. The second one is the spatially and temporally complete (STC) MODIS Snow-Covered Area and Grain size (MODSCAG) and MODIS Dust and Radiative Forcing in Snow (MODDRFS) product
(we hereafter refer to as STC-MODSCAG/STC-MODDFRS). The third one is the Snow Property Inversion From Remote Sensing (SPIReS) product. These two products provide $f_{\mathrm{sno}}$, $\alpha_{\mathrm{sno}}$, $S_{\mathrm{sno}}$, and $R_{\mathrm{sno}}$ at around 10:30 am local solar time and represent $\alpha_{\mathrm{sno}}$ (i.e. excluding soil and vegetation portions of the observed pixel). STC-MODSCAG first estimates $f_{\mathrm{sno}}$ and $S_{\mathrm{sno}}$ based on the spectral unmixing and physically-based snow radiative transfer models (Painter et al., 2009). STC-MODDRFS then uses $S_{\mathrm{sno}}$ to calculate the $\alpha_{\mathrm{sno}}$ of the clean snow with difference between clean and dirty (observed) snow for computing $R_{\mathrm{sno}}$ (Painter
et al., 2012). SPIReS adopts a physically-based approach without empirical assumptions to simultaneously estimate $f_{\mathrm{sno}}$, $\alpha_{\mathrm{sno}}$, $S_{\mathrm{sno}}$, and $R_{\mathrm{sno}}$ (Bair et al., 2021c). Both STC-MODSCAG/STC-MODDRFS and SPIReS are interpolated and smoothed to reduce the effects of data noise, cloud contamination and sun-sensor geometry (Bair et al., 2021c; Dozier et al., 2008; Rittger et al., 2020). Both of the $f_{\mathrm{sno}}$ products show good performance with the basin-wide root mean square error (RMSE) values of 6.5% and 6.7% against airborne lidar datasets (Stillinger et al., 2022). Initial validation against field measurements for $S_{\mathrm{sno}}$ at
a single site for the original MODSCAG shows a 51 µm mean absolute error for a clear sky day (Painter et al., 2009). The gap filled MODSCAG/MODDRFS at three sites in the WUS has an accuracy (RMSE) of 118 µm for $S_{\mathrm{sno}}$ and 0.0036 for $R_{\mathrm{sno}}$ (Bair et al., 2019) considering both clear and cloud days. SPIReS has a $\alpha_{\mathrm{sno}}$ RMSE of 4.6% against the 3-year field measurements at Mammoth Mountain, CA (Bair et al., 2021c), nearly identical to the reported accuracy of 4.8% RMSE for STC-MODDRFS agaist the field measurements at the same site (Bair et al., 2019). Note that there is an underestimation of $f_{\mathrm{sno}}$ in the northern
WUS region in winter occur because of a known issue in current versions of STC-MODSCAG (https://nsidc.org/reports/snow-today?title=6). Specifically, MOD09GA surface reflectance processed to produce STC-MODSCAG at the Jet Propulsion Laboratory (JPL) is not processed when SZA is larger than 67.5°. This issue is being resolved during the transfer of processing during 2022 to 2023 from JPL to the National Snow and Ice Data Center Distributed Active Archive. We conservatively excluded data north of 42° in latitude during the winter in our comparisons in Section 3.1.

Two data assimilation SWE and $D_{\mathrm{sno}}$ products from 2001-2019 were used to compare with ELM (Table 1). The first one is the University of Arizona (UA) daily snow product version 1 with the spatial resolution of 4 km over the Conterminous US (Zeng et al., 2018). This product was generated by fully utilizing the field measurements from multiple in situ networks including SNOTEL constrained by the gridded precipitation and temperature data in the 4 km Parameter-elevation Regressions on
Independent Slopes Model (PRISM). A series of algorithm robustness tests and independent accuracy evaluations against





remote sensing and airborne lidar measurements showed that the UA product is reliable as a reference snowpack dataset (Zeng et al., 2018). The second one is the SNOw Data Assimilation System (SNODAS) daily product with 1 km spatial resolution developed by the NOAA National Weather Service's National Operational Hydrologic Remote Sensing Center (Center, 2004). SNODAS uses a physically consistent modeling and data assimilation framework to integrate physically-based model
estimates and multi-source snow data from satellite remote sensing, airborne-based observations, and in situ measurements including SNOTEL. SNODAS has shown a similar performance as UA (Zeng et al., 2018). The SNODAS product is available from October, 2003 and thus only the data from 2004-2019 were used in the study. UA and SNODAS both assimilate the SNOTEL observations in their models directly, so better performance relative to those observations is expected, while the ELM simulations are not constrained by the SNOTEL data.


**Table 1. Summary of the in situ, remote sensing and data assimilation datasets used in the study. These datasets provide different snow properties variables, and snow cover fraction in both STC-MODSCAG/STC-MODDRFS and SPIReS was used to derive snow phenology metrics.**

| Product Type | Product name | Snow property | Spatial resolution | Temporal resolution | Period | Reference |
|---|---|---|---|---|---|---|
| In situ | BCQC SNOTEL | Snow water equivalent (SWE) | Site-level | daily | 2001-2019 | (Sun et al., 2019; Yan et al., 2018) |
| Remote sensing | MODIS MCD43A3 | Surface albedo ($\alpha_{sur}$) | 463 m | daily | 2001-2019 | (Schaaf et al., 2002) |
| | STC-MODSCAG /STC-MODDRFS | Snow cover fraction ($f_{sno}$) Snow albedo ($\alpha_{sno}$) Snow albedo reduction ($R_{sno}$) Snow grain size ($S_{sno}$) | 463 m | daily | 2001-2019 | (Rittger et al., 2020) |
| | SPIReS | Snow cover fraction ($f_{sno}$) Snow albedo ($\alpha_{sno}$) Snow albedo reduction ($R_{sno}$) Snow grain size ($S_{sno}$) | 463 m | daily | 2001-2019 | (Bair et al., 2021c) |
| Data assimilation | UA | Snow water equivalent (SWE) Snow depth ($D_{sno}$) | 4 km | daily | 2001-2019 | (Broxton et al., 2019; Zeng et al., 2018) |
| | SNODAS | Snow water equivalent (SWE) Snow depth ($D_{sno}$) | 1 km | daily | 2004-2019 | (Center, 2004) |

**2.4 Snow phenology extraction and data processing**

Time series of $f_{sno}$ from ELM and two remote sensing snow products (i.e., STC-MODSCAG and SPIReS) were used to extract the snow phenology (Figure 2). First, based on the observed seasonal cycle of snow cover over the Northern



Hemisphere (Peng et al., 2013), the snow accumulation and snowmelt seasons are defined as the periods from September to January and from February to August, respectively. Next, four snow timing dates and one duration metric were retrieved

from ELM and remote sensing products that include: (1) snow accumulation onset date (Accumulation_onset_date), (2) snow cover depletion onset date (Depletion_onset_date), (3) snow cover depletion midpoint date (Midpoint_date), (4) snow end date (End_date) and (5) snow duration days (Duration). Following (Peng et al., 2013), Accumulation_onset_date for year $t$ is defined as the first continuous five days with $f_{sno}$>0.05 during the snow accumulation season from September (year $t$-1) to January (year $t$), and End_date is defined as the last continuous five days with $f_{sno}$>0.05 during the snowmelt season of

the year t, to avoid the interference of ephemeral snow. Note that using different thresholds (e.g., 0.00, 0.03, 0.05, 0.10, 0.15) of $f_{sno}$ to defining Accumulation_onset_date and End_date can lead to different date estimates but the same conclusions, which are not shown in the paper. Duration was calculated as the number of days between Accumulation_onset date and End_date. Depletion_onset_date and Midpoint_date were determined by fitting the $f_{sno}$ time series during the snowmelt season using the sigmoid function (Anttila et al., 2018; Böttcher et al., 2014; Kouki et al., 2019):

$$f_{sno}^{DOY} = a + \frac{b}{1+e^{c \cdot (DOY-d)}} \tag{1}$$

where DOY is day of year, and a, b, c and d are the fitted parameters. Specifically, the nonlinear Least Squares method was used to fit a sigmoid function. Following Anttila et al.(2018), Depletion_onset_date is defined as the date when the fitted sigmoid curve reaches 99% of its variation range, and Depletion_midpoint_date is defined as the date at the midpoint of the curve change (Figure 2). To reduce the impacts of noise, the retrievals at the individual pixels for a specific year was deemed

as unsuccessful when (1) the $f_{sno}$ difference at the start and end date of snowmelt season is smaller than 0.05; and (2) for the sigmoid fitting, the coefficient of determination ($R^2$) between observed and fitted $f_{sno}$ is smaller than 0.95 and RMSE is larger than 0.2. Only the pixels with successful retrievals of snow timing metrics for at least 10 years were used in the subsequent analysis.





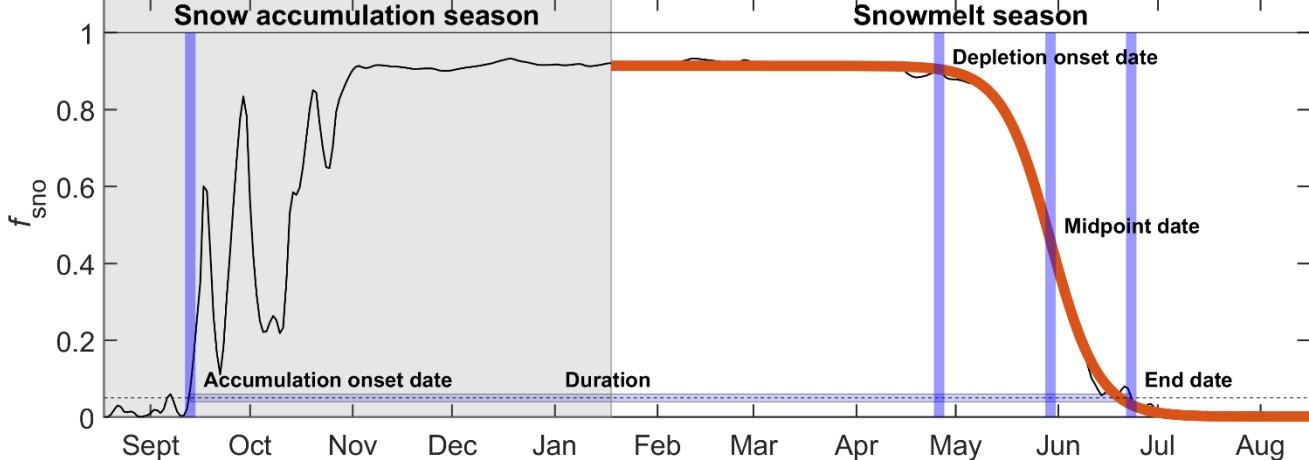


**Figure 2: Time series of $f_{sno}$ and sigmoid curve fitting at a typical pixel, represented by black and red lines, respectively. The blue lines indicate four phenology dates and one duration, and the shaded area shows the snow accumulation season.**

MODIS data and ELM outputs were adjusted for temporal consistency and to unify the variable definitions. MCD43 only

provides black-sky and white-sky albedo, and thus the ELM-derived ratio of diffuse to total solar radiation was used as a weighting factor to calculate $\alpha_{sur}$ for the blue sky. For ELM, the average values of $\alpha_{sur}$ from 11:30 am to 12:30 pm local solar time were calculated to match the time of MODIS MCD43 product, and those of $f_{sno}, \alpha_{sno}, S_{sno}$, and $R_{sno}$ from 10:00 am to 11:00 am local solar time were calculated for ELM to match the time of STC-MODSCAG/STC-MODDRFS and SPIReS.

The snow timing metrics and snow variables in the remote sensing and data assimilation products (Table 1) were aggregated to 0.125° using the area-weighted average method. They were temporally upscaled to seasonal, annual and multi-year average scales. For a specific year, only the pixels with $f_{sno}>0$ were used to calculate the regional average values for $\alpha_{sur}, f_{sno}, \alpha_{sno}, S_{sno},$ and $R_{sno}$, SWE and $D_{sno}$ using the area-weighted average method.

**2.5 Evaluation methods**

Using the field measurements, remote sensing products, and data assimilation products as the reference, the spatio-temporal distributions of ELM snow outputs were evaluated. For spatial correlation, multiple statistical metrics were calculated for the multi-year average seasonal ELM outputs: correlation coefficient (R), Bias, relative Bias (rBias, calculated as the ratio of Bias to the average value), root mean square deviations (RMSD), and relative RMSD (rRMSD, calculated as the ratio of RMSD to

the average value). This study mainly focused on winter (DJF) and spring (MAM) in the analysis, and there is little or no snow cover for the WUS in Summer (JJA) and Autumn (SON) in the ELM simulations (Figure S1). For the temporal correlation, R





between ELM and the reference datasets was calculated only for the grids where there are at least 10 snow covered days for one year excluding highly ephemeral snow.

The long-term trends of snow variables over the whole WUS were detected using the non-parametric Mann–Kendall (MK) test. However, the MK test showed that there is no significant increasing or decreasing trend (p-value > 0.05) for all the snow variables, and thus the corresponding results are not included in the paper. The interannual variabilities (IAVs), defined as the standard deviation of the annual values, were calculated to evaluate whether ELM can capture the interannual variations of snow processes. In addition, the distributions of snow variables along the elevation gradients and forest cover for winter and 260 spring were also analyzed.

## 3 Results

### 3.1 Snow properties

#### 3.1.1 Snow cover fraction

The ELM simulated $f_{sno}$ has heterogeneous spatial patterns in the WUS for both winter and spring (Figure 3a-b). The regional 265 average $f_{sno}$ is 0.41 and 0.15, respectively for spring and winter. Overall, ELM also shows similar spatial patterns with both STC-MODSCAG and SPIReS for all the seasons (Figure S1). STC-MODSCAG underestimates $f_{sno}$ over the northern regions in winter due to the known issues (Figure S1, see Section 2.3 for details). When excluding December and January with larger SZAs, STC-MODSCAG shows similar spatial distribution as SPIReS for February (Figure S2). In spring, compared to STC-MODSCAG, ELM underestimates $f_{sno}$ over the western mountains in spring (Figure 3d). Compared to SPIReS, ELM has an 270 overestimation over most regions in winter but performs well in spring (Figure 3g-h). Overall ELM has a high spatial correlation to both STC-MODSCAG and SPIReS (Table 2). For temporal correlation, ELM has a moderate correlation in the mountainous areas with both STC-MODSCAG and SPIReS in winter (Figure 3e,i), but has a relatively high correlation with them in spring (Figure 3f,j).

ELM well reproduces the interannual variabilities and elevation gradients of $f_{sno}$ (Figure 4 and S3). The IAV values are 0.055 and 0.049, respectively for ELM and SPIReS in winter, while they have closer values of 0.027, 0.029 and 0.030, respectively for ELM, STC-MODSCAG and SPIReS in spring (Figure 4a-b). ELM underestimates regional average $f_{sno}$ in spring, and is overall consistent with STC-MODSCAG and SPIReS in terms of magnitude and IAVs. As the elevation increases, $f_{sno}$ values in all three datasets become higher for both winter and spring (Figure 4c-d). At relatively low elevation, the $f_{sno}$ distributions 280 in ELM are broader than those of SPIReS in winter, while the three datasets have more consistent elevation gradients in spring. Overall, when forest cover is higher, ELM show larger differences with SPIRES for spring and STC-MODSCAG for winter (Figure 4e-f). Same conclusions can be drawn for the regions below 42° in latitude (Figure S3). Considering the uncertainties



of the remote sensing retrievals, the ELM regional average $f_{sno}$ is within the range of STC-MODSCAG and SPIReS (Figure 5a-b and S4).


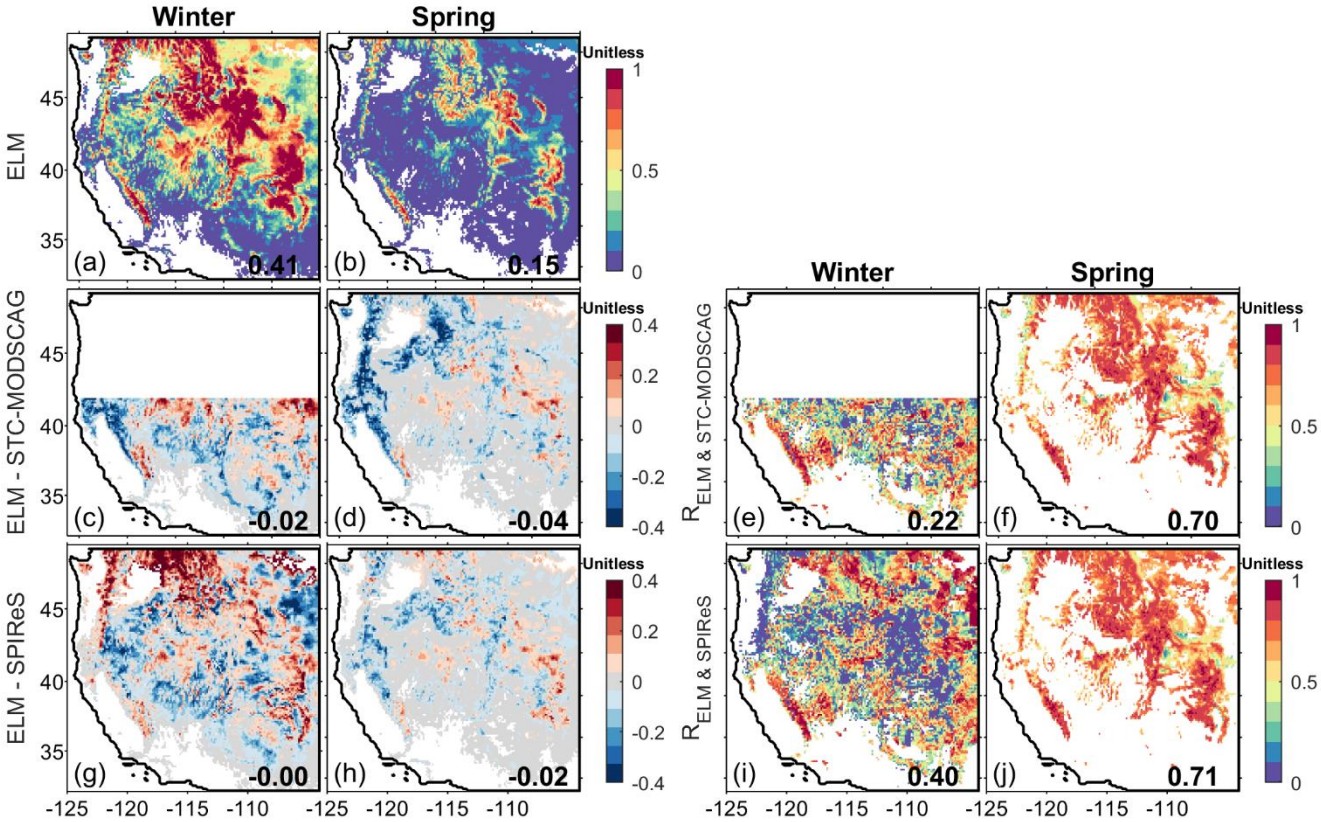

**Figure 3: Spatial distributions of (a,b) $f_{sno}$ in ELM and (c,d,g,h) the $f_{sno}$ difference between ELM and two remote sensing products (i.e., STC-MODSCAG and SPIReS) and (e,f,i,j) their temporal correlations (Rs) for different seasons: (a,c,e,g,i) winter and (b,d,f,h,j) spring. In all panels, regions with no snow cover are masked with white color. The area-weighted average values are labelled in each**
**panel.**





**Figure 4: (a,b)** Time series of regional average values, **(c,d)** elevation gradients, and **(e,f)** change with forest cover of $f_{sno}$ in ELM (green), STC-MODSCAG (red), and SPIReS (blue) over the WUS. Panels (a,c,e) are for winter and panels (b,d,f) are for spring. In panels (c-f), the white dots represent the average values.





**Figure 5: The area-weighted average (a,b) $f_{sno}$, (c,d) $S_{sno}$ and (e,f) $R_{sno}$ for (a,c,e) winter and (b,d,f) spring of ELM (green), STC-MODSCAG/STC-MODDRFS (red) and SPIReS (blue) over the WUS. The bar width represents the uncertainty bounds of STC-MODSCAG/STC-MODDRFS and SPIReS from (Bair et al., 2021a).**







**Table 2. Evaluation of snow properties in ELM against two remote sensing products (STC-MODSCAG/STC-MODDRFS and SPIReS) and two data assimilation products (UA and SNODAS) for winter and spring. The statistical metrics were calculated using the data over the WUS, except that those against STC-MODSCAG/STC-MODDRFS in winter were calculated using the data over the WUS regions below 42° in latitude.**

| Variables | Products | Winter | | | | | Spring | | | | |
|---|---|---|---|---|---|---|---|---|---|---|---|
| | | R | Bias | rBias (%) | RMSD | rRMSD(%) | R | Bias | rBias (%) | RMSD | rRMSD(%) |
| $f_{sno}$ | STC-MODSCAG | 0.91 | -0.03 | -10.4 | 0.13 | 39.5 | 0.90 | -0.04 | -22.1 | 0.11 | 57.8 |
| | SPIReS | 0.86 | 0.00 | -1.0 | 0.16 | 39.1 | 0.94 | -0.02 | -11.7 | 0.08 | 46.6 |
| $\alpha_{sur}$ | MCD43 | 0.77 | -0.014 | -4.2 | 0.097 | 30.1 | 0.71 | 0.004 | 2.3 | 0.056 | 29.6 |
| $\alpha_{sno}$ | STC-MODDRFS | -0.09 | -0.15 | -19.3 | 0.18 | 22.2 | -0.27 | -0.11 | -14.7 | 0.13 | 17.6 |
| | SPIReS | 0.15 | -0.13 | -16.2 | 0.16 | 19.5 | -0.09 | -0.08 | -11.4 | 0.11 | 14.8 |
| $S_{sno}$ (μm) | STC-MODSCAG | -0.15 | 78.2 | 37.7 | 159.3 | 76.9 | 0.02 | -71.6 | -17.2 | 226.5 | 54.4 |
| | SPIReS | 0.16 | 93.9 | 50.6 | 120.6 | 65.0 | 0.18 | 31.6 | 10.1 | 128.2 | 40.9 |
| $R_{sno}$ | STC-MODDRFS | 0.58 | -0.007 | -77.7 | 0.011 | 126.7 | 0.50 | 0.000 | -8.7 | 0.006 | 153.1 |
| | SPIReS | 0.10 | -0.002 | -26.4 | 0.014 | 170.0 | 0.63 | -0.007 | -66.3 | 0.013 | 118.8 |
| SWE(mm) | UA | 0.91 | -13.8 | -27.8 | 37.1 | 75.1 | 0.90 | -20.7 | -35.9 | 62.9 | 108.9 |
| | SNODAS | 0.90 | -10.2 | -22.2 | 36.7 | 80.1 | 0.87 | -20.4 | -35.5 | 71.5 | 124.5 |
| $D_{sno}$(mm) | UA | 0.92 | -39.9 | -21.6 | 119.2 | 64.5 | 0.91 | -70.0 | -43.2 | 172.9 | 106.8 |
| | SNODAS | 0.90 | -48.1 | -24.9 | 138.9 | 72.0 | 0.87 | -85.7 | -48.2 | 228.8 | 128.9 |

### 3.1.2 Surface albedo and snow albedo

Overall the ELM simulated $\alpha_{sur}$ over snow cover regions shows similar spatio-temporal distribution with MCD43 for both winter and spring (Figure 6-7). Compared to MCD43, ELM overestimates $\alpha_{sur}$ over Sierra Nevada and Rocky Mountains in winter, possibly due to the bias in snow cover (Figure 3c-d). The mean biases of ELM are -0.01 and 0.00, respectively for winter and spring. The spatial R values between ELM and MCD43 are 0.77 and 0.71, respectively for winter and spring (Table 2). ELM shows a low temporal correlation to MCD43 over most regions in winter, but has a relatively higher temporal correlation in spring especially over the mountain areas and northern regions (Figure 6e-f). ELM also has similar interannual variability especially in winter (Figure 7a-b), similar elevation gradient (Figure 7c-d) and similar distributions under different





forest cover (Figure 7e-f) with MCD43. As $f_{sno}$ increases, $\alpha_{sur}$ in both ELM and MCD43 increases and ELM and MCD43 have

similar $\alpha_{sur}$ distributions for different elevation intervals (Figure 7g-h).

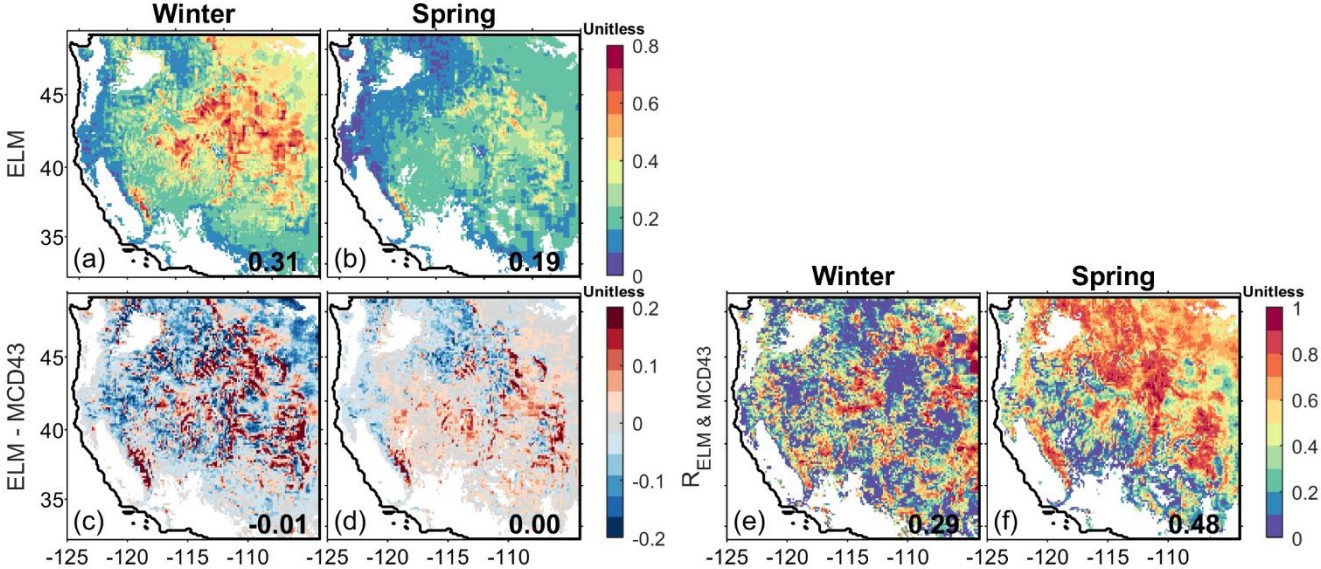

**Figure 6: Spatial distributions of (a,b) $\alpha_{sur}$ in ELM and (c,d) the $\alpha_{sur}$ difference between ELM and MCD43 and (e,f) their temporal**
**correlations (Rs) for different seasons: (a,c,e) winter and (b,d,f) spring. In all panels, the regions with no snow cover are masked**
**with white color. The area-weighted average values are labelled in each figure.**





**Figure 7:** (a,b) Time series of regional average values, (c,d) elevation gradients, (e-f) change with forest cover and (g,h) statistical distributions of $\alpha_{sur}$ under different snow cover conditions in ELM (green) and MCD43 (red) for different seasons: (a,c,e,g) winter and (b,d,f,h) spring over the WUS. The IAV values of different datasets are shown in (a,b). In panels (c-h), the white dots represent the average values.





For $\alpha_{sno}$, ELM overall shows good consistencies with STC-MODDRFS and SPIReS over mountainous regions, but has an underestimation over other regions (Figure 8). Against STC-MODDRFS, the mean biases of ELM are -0.08 for winter over

the WUS regions below 42° in latitude and -0.11 for spring over the WUS. Against SPIReS, the mean biases of ELM are -0.13 and -0.08, respectively for winter and spring. The spatial R values between ELM and two remote sensing products are lower than 0.30 (Table 2). ELM shows a low temporal correlation to two remote sensing products over most regions, and has a relatively higher temporal correlation over the Rocky Mountains (Figure 8e-f). Larger inconsistencies between ELM and two remote sensing products are founded in terms of interannual variations, elevation gradients and change with forest cover

(Figure 9 and S5).

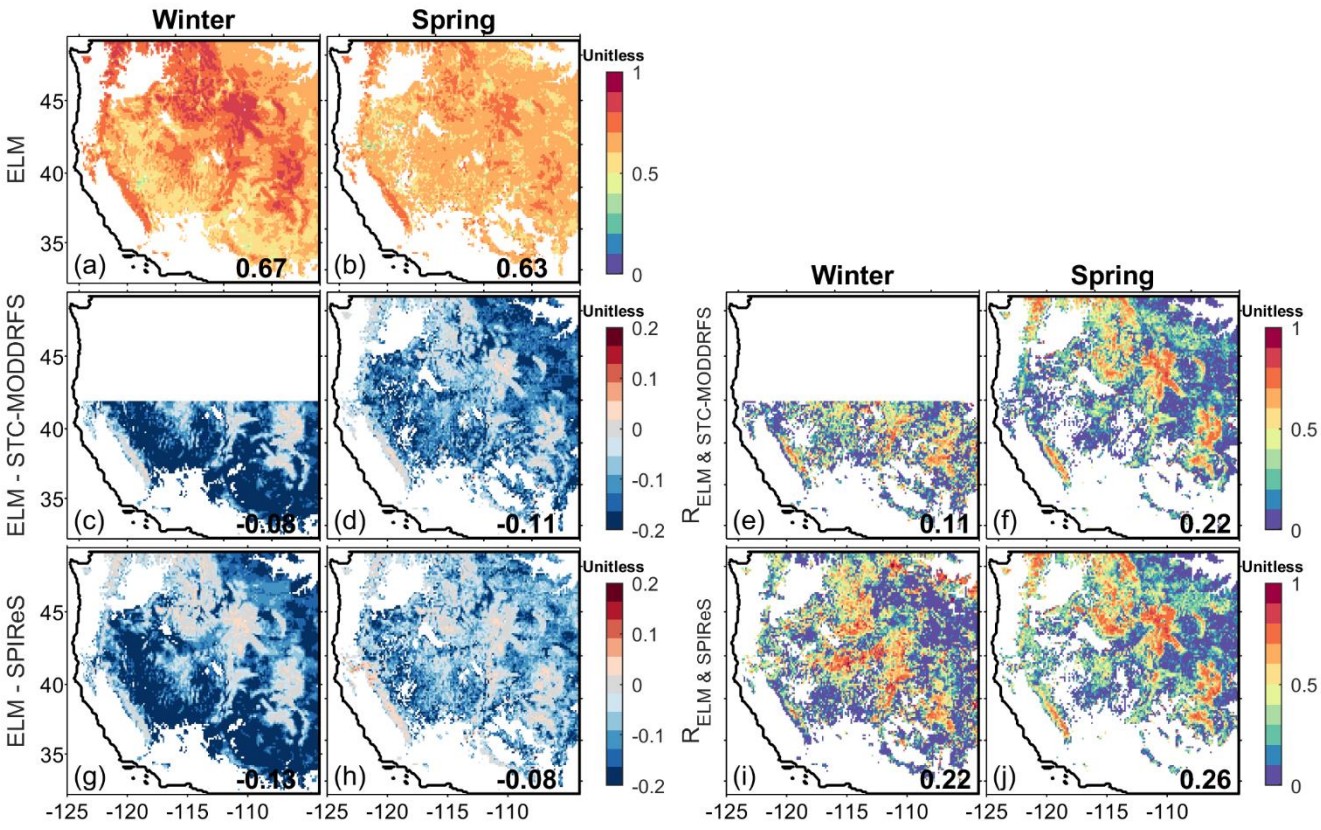

**Figure 8: Spatial distributions of (a,b) $\alpha_{sno}$ in ELM and (c,d,g,h) the $\alpha_{sno}$ difference between ELM and two remote sensing products (i.e., STC-MODDRFS and SPIReS) and (e,f,i,j) their temporal correlations (Rs) for different seasons:**
**(a,c,e,g,i) winter and (b,d,f,h,j) spring. In all panels, regions with no snow cover are masked with white color. The area-weighted average values are labelled in each panel.**





**Figure 9:** (a,b) Time series of regional average values, (c,d) elevation gradients, and (e,f) change with forest cover of $\alpha_{sno}$ in ELM (green), STC-MODSCAG (red), and SPIReS (blue) over the WUS. Panels (a,c,e) are for winter and panels (b,d,f) are for spring. In panels (c-f), the white dots represent the average values.

### 3.1.3 Snow grain size and snow albedo reduction

There are large differences in the magnitudes and spatio-temporal patterns of $S_{sno}$ between ELM, STC-MODSCAG/SPIReS (Figure 10 and 11). ELM has larger $S_{sno}$ in spring than in winter (Figure 10a-b), with large negative biases over the western mountains and positive biases over the central and eastern regions compared to STC-MODSCAG with the mean biases of -71.6 μm for spring (Figure 10c-d). ELM has positive biases over most regions compared to SPIReS, with the mean bias of 93.9 μm and 31.6 μm, for winter and spring, respectively (Figure 10g-h). $S_{sno}$ in ELM has a poor spatial correlation to the two
MODIS products for both winter and spring (Table 2). ELM has varying temporal correlations with STC-MODSCAG and





SPIReS for both seasons with a mean value of around 0.3 (Figure 10 e-f and i-j). ELM has a similar interannual variability to SPIReS (Figure S6a-b and S7a-b). As the elevation increases, ELM and SPIReS have decreasing $S_{sno}$, in winter, but there is no obvious and comparable pattern along the elevation in spring (Figure S6c-d and S7c-d). As forest cover increases, the three data show larger differences for spring (Figure S6f and S7f). Considering the uncertainties of $S_{sno}$ in the remote sensing

products, the regional average $S_{sno}$ is within the range between STC-MODSCAG and SPIReS (Figure 5c-d and S4).

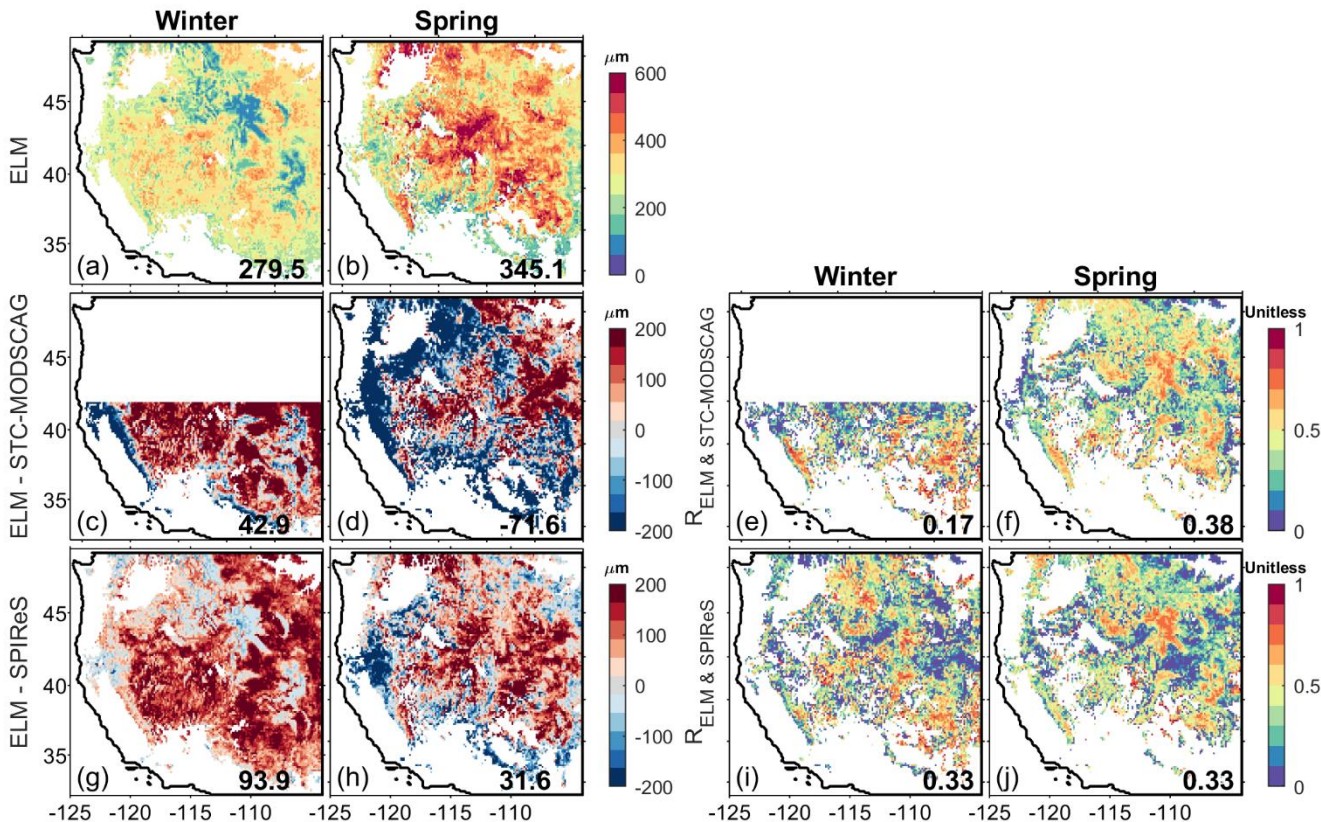

**Figure 10: Spatial distributions of (a,b) $S_{sno}$ in ELM and (c,d,g,h) the $S_{sno}$ difference between ELM and two remote sensing products (i.e., STC-MODSCAG and SPIReS) and (e,f,i,j) their temporal correlations (Rs) for different seasons: (a,c,e,g,i) winter and (b,d,f,h,j) spring. In all panels, regions with no snow cover are masked with white color. The area-**
**weighted average values are labelled in each panel.**

There are also large spatial biases and low temporal correlations of $R_{sno}$ between ELM, STC-MODDRFS and SPIReS (Figure 11 and S4). In ELM, $R_{sno}$ shows extremely high values in the northeastern corner for winter (Figure 11a), due to the large aerosol deposition in the aerosol deposition data (see Section 2.2). Apart from the northeastern corner, ELM is more similar to SPIReS in winter (Figure 11c-g). For spring, ELM is more similar to STC-MODSCAG, and has large negative biases

relative to SPIReS (Figure 11d-h). ELM has higher temporal correlations with both remote sensing products in winter than spring, and shows higher correlations with SPIReS than STC-MODDRFS in spring (Figure 11 e-f and i-j). For interannual





variability, ELM is more identical to STC-MODSCAG in spring (Figure S8a-b and S9a-b) than SPIRES. However, note that ELM simulations in the study used climatological monthly aerosol deposition data, so they are not comparable to the remote sensing data in any specific year. In spring, $R_{sno}$ in all the three datasets shows an increasing trend with elevation (Figure S8d and S9d). All the three data show larger differences across different forest cover (Figure S8e-f and S9e-f). Overall, $R_{sno}$ is within the uncertainty ranges of STC-MODSCAG and SPIReS (Figure 5e-f and S4).

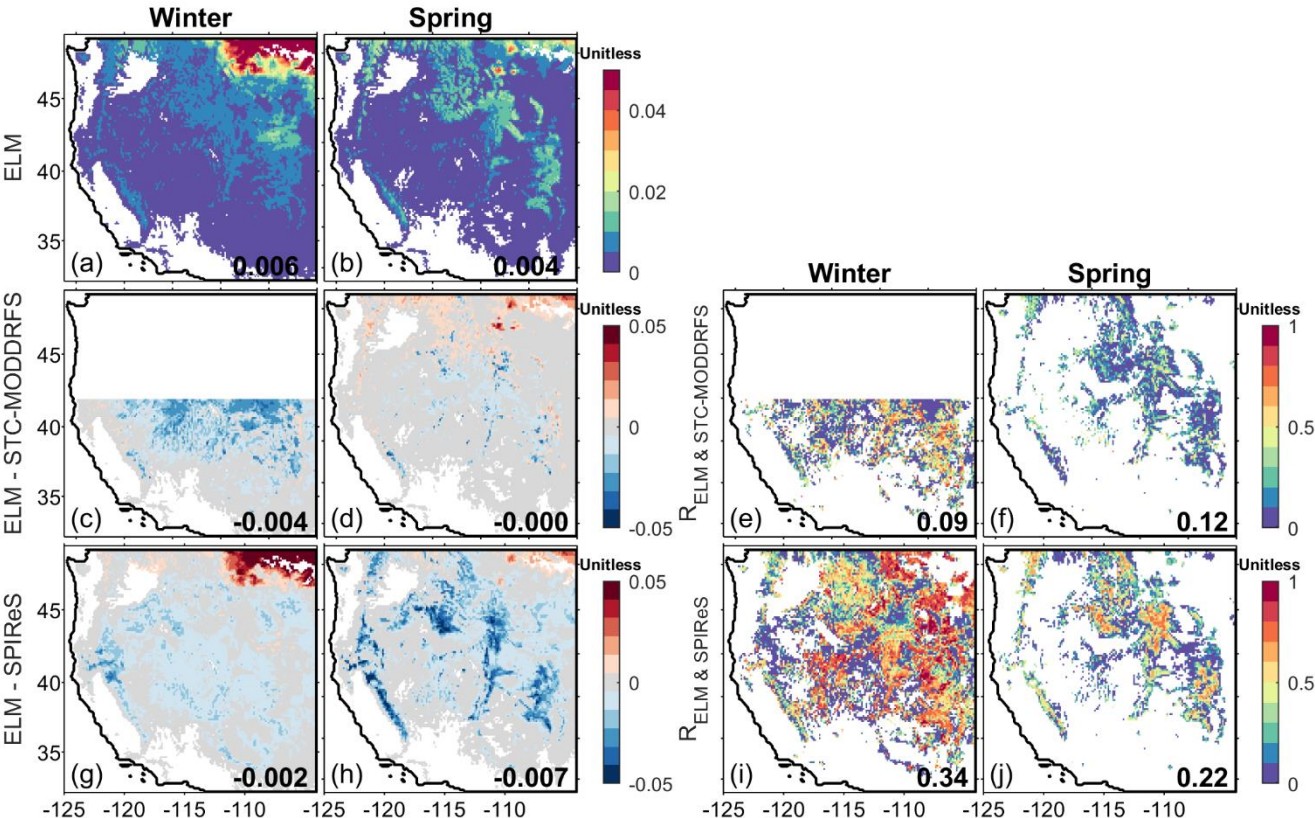

**Figure 11: Spatial distributions of (a,b) $R_{sno}$ in ELM and (c,d,g,h) the $R_{sno}$ difference between ELM and two remote sensing products (i.e., STC-MODDRFS and SPIReS) and (e,f,i,j) their temporal correlations (Rs) for different seasons: (a,c,e,g,i) winter and (b,d,f,h,j) spring. In all panels, regions with no snow cover are masked with white color. The area-weighted average values are labelled in each panel.**

### 3.1.4 Snow water equivalent and snow depth

ELM shows higher SWE values over the mountainous areas (Figure 12a-b), but also has larger underestimations over the mountainous areas, compared to both UA and SNODAS in both winter and spring (Figure 12c-d,g-h). Against UA and SNODAS, ELM has a mean bias of -20.7 mm (35.9%) and -20.4 mm (-35.5%), respectively in spring, while those in winter are -13.8 mm (-27.8%) and -10.2 mm (-22.2%), respectively. Overall ELM has a high spatial similarity with both UA and





SNODAS, and ELM has higher spatial consistency with UA than SNODAS in spring (Table 2). For temporal correlation
(Figure 12e-f and i-j), ELM has high mean R values of 0.64 and 0.65 for winter and spring, compared to UA, and the R values
       are 0.53 and 0.54, respectively compared to SNODAS. ELM captures the interannual variabilities and elevation gradients of
       SWE well, but some underestimations of the regional average values are observed (Figure 13a-d). In winter, ELM has similar
       IAV values to UA and SNODAS, but has a lower value of 11.7 mm compared to UA (16.7 mm) and SNODAS (18.1 mm) in
       spring. Overall, ELM shows larger differences from UA and SNODAS, when there is a higher forest cover, especially for
spring (Figure 13e-f). $D_{sno}$ shows very similar results to SWE (Figure S10-S11).

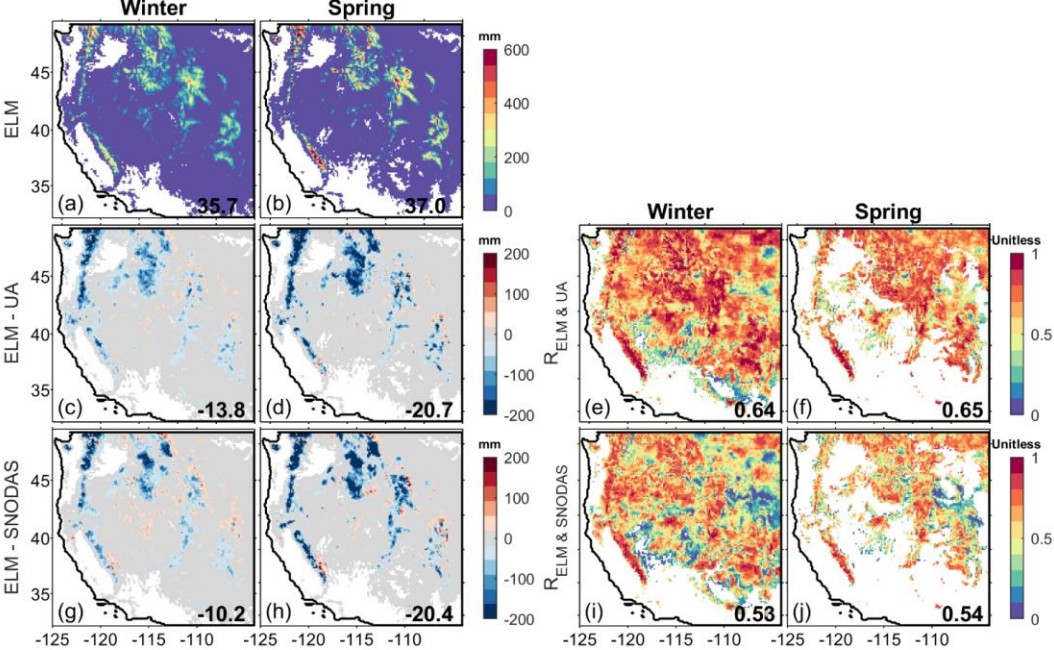

**Figure 12: Spatial distributions of (a,b) SWE in ELM and (c,d,g,h) the SWE difference between ELM and two data assimilation products (i.e., UA and SNODAS) and (e,f,i,j) their temporal correlations (Rs) for different seasons: (a,c,e,g,i) winter and (b,d,f,h,j) spring. In all panels, regions with no snow cover are masked with white color. The area-**
**weighted average values are labelled in each panel.**







**Figure 13: (a,b) Time series of regional average values, (c,d) elevation gradients, and (e,f) change with forest cover of SWE in ELM (green), UA (red), and SNODAS (blue) over the WUS. Panels (a,c,e) are for winter and panels (b,d,f) are for spring. In panels (c-f), the white dots represent the average values.**

Compared to SNOTEL, UA presents a high correlation across sites (Figure 14), with the mean R values are 0.69. The mean RMSE of ELM is 189.6 mm, the Cascades Range shows larger RMSE values than other regions. ELM underestimates SWE nearly across all sites, with the mean biases of -122.7 mm. The biases of the meteorological forcing in NLDAS-2 and the spatial-scale mismatch between the point-scale SNOTEL and the grid-level ELM simulations can contribute to uncertainty in the comparison.





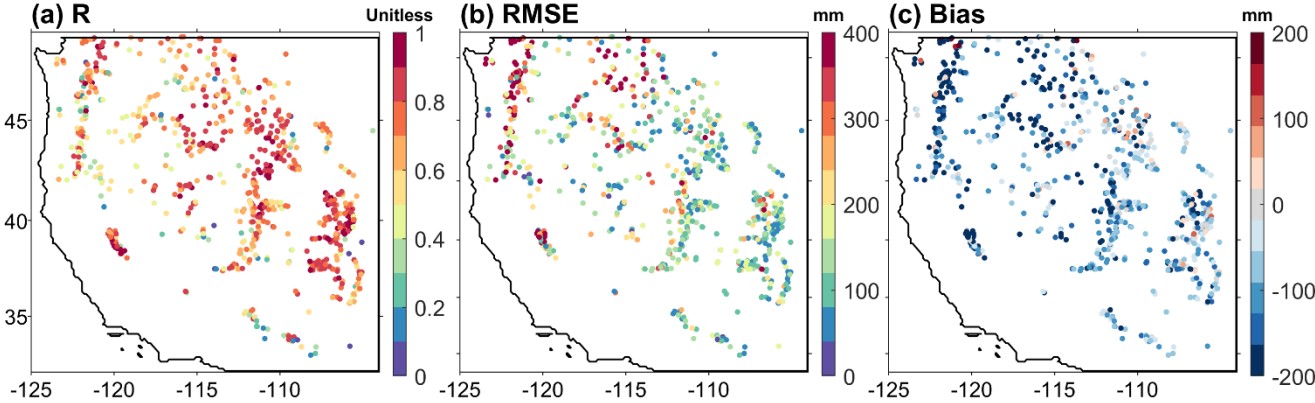


**Figure 14. Spatial distribution of statistical metrics of ELM performance against the field SNOTEL data: (a) R, (b) Bias, and (c) RMSE.**

**3.2 Snow phenology**

ELM well reproduces the snow phenology, compared to two remote sensing products (Figure 15-16). As expected, over
mountainous areas, ELM shows earlier snow onset, later depletion and thus longer snow duration compared to flat and generally lower elevation areas (first column of Figure 15). Compared to STC-MODSCAG and SPIReS (second and third columns of Figure 15), ELM shows later Accumulation_onset_date over the whole WUS with a mean bias of +17.3 and +12.4 days, respectively, which may be caused by the bias in the meteorological forcing data of NLDAS-2 and the simple parameterizations of the partitioning of precipitation into rainfall or snowfall; and has later Depletion_onset_date, but earlier
Midpoint_date and End_date. For instance, ELM melts off earlier with a mean bias of -35.5 and –26.8 days, respectively than STC-MODSCAG and SPIReS, suggesting that ELM has higher snowmelt rate. Thus ELM has a short snow duration with a mean bias of -52.9 and -39.5 days, respectively compared to the two remote sensing products. The large biases exist in the western mountains for End_date (Figure 15k-l and n-o). Overall snow phenology in ELM has a high spatial correlation with that of the remote sensing products (Table 3). Although ELM overestimates Accumulation_onset_date and
Depletion_onset_date and underestimates Midpoint_date, End_date and Duration, ELM well captures the IAVs of all five snow phenology metrics (first column of Figure 16), As the elevation increases, Accumulation_onset_date decreases but the other four metrics increases for all the three datasets (second column of Figure 16). ELM also has similar magnitudes and distributions for each elevation interval compared to the remote sensing products, while the three data show larger and larger differences with the increase of forest cover (third column of Figure 16).






**Figure 15: Spatial distributions of (a,d,g,j,m) snow timing and (b-c,e-f,h-i,k-l,n-o) the snow timing difference between ELM and two remote sensing products ((i.e., STC-MODSCAG and SPIReS). Five snow timing metrics are included: (a-c) Accumulation_onset_date, (d-f) Depletion_onset_date, (g-i) Midpoint_date, (j-l) End_date, and (m-o) Duration. The regions with no successful retrievals of snow timing are masked with white color. The area-weighted average values are labelled in each figure.**





**Figure 16: (a,d,g,j,m) Time series of regional average values, (b,e,h,k,n) elevation gradients and (c,f,i,l,o) change with forest cover of snow timing in ELM and two remote sensing products (i.e., STC-MODSCAG and SPIReS) for different metrics: (a-c) Accumulation_onset_date, (d-f) Depletion_onset_date, (g-i) Midpoint_date, (j-l) End_date, and (m-o) Duration over the WUS. The IAV values of different data are shown in (a,d,g,j,m). In panels (b-c,e-f,h-i,k-l,n-o), the white dots represent the average values.**





**Table 3. Evaluation of snow phenology in ELM against STC-MODSCAG and SPIReS.**

| Products | Variables | R | Bias | rBias(%) | RMSD | rRMSD(%) |
|---|---|---|---|---|---|---|
| STC-MODSCAG | Accumulation_onset_date | 0.83 | 17.3 | 5.6 | 22.0 | 7.1 |
| | Depletion_onset_date | 0.77 | 6.8 | 9.4 | 15.6 | 21.6 |
| | Midpoint_date | 0.91 | -9.2 | -8.1 | 15.2 | 13.4 |
| | End_date | 0.81 | -35.5 | -32.1 | 42.9 | 38.9 |
| | Duration | 0.84 | -52.9 | -30.0 | 63.6 | 36.1 |
| SPIReS | Accumulation_onset_date | 0.86 | 12.4 | 3.9 | 14.6 | 4.6 |
| | Depletion_onset_date | 0.82 | 10.6 | 15.7 | 16.0 | 23.7 |
| | Midpoint_date | 0.93 | -5.7 | -5.3 | 12.6 | 11.7 |
| | End_date | 0.89 | -26.8 | -26.4 | 32.2 | 31.7 |
| | Duration | 0.90 | -39.5 | -25.0 | 45.2 | 28.5 |


## 4 Discussion

The evaluation results suggest an overall good performance of ELM in simulating snow properties, while some biases and uncertainties still exist, especially over mountainous areas with dense forest cover. Compared to the remote sensing products, ELM well reproduces the spatio-temporal pattern, interannual variabilities and elevation gradients of $f_{sno}$ and $\alpha_{sur}$ (Figure 3-6),

but large biases exist in Rocky Mountains and Sierra Nevada for $\alpha_{sur}$ (Figure 3 and 5). There are still large spatio-temporal inconsistencies of $S_{sno}$ and $R_{sno}$ among ELM, STC-MODSCAG and SPIReS (Figure 10-11 and S6-S9). The underestimation of SWE and snow depth by ELM is comparable to the reported results based on CLM4 (Toure et al., 2018; Toure et al., 2016). The NLDAS-2 data used in the ELM simulations has large negative precipitation biases and high air temperature uncertainties over high-elevation terrain compared to both field measurements and PRISM over the WUS (Henn et al., 2018; O'Neill et al.,

2021; Pan et al., 2003; Schreiner-McGraw and Ajami, 2022), which can partly explain the negative SWE bias in ELM. Besides, a 0.125° grid may have high sub-grid variabilities of snow especially in mountainous areas (Meromy et al., 2013) and SNOTEL stations in mountains located on flat surface may not capture the sub-grid spatial variabilities (Toure et al., 2016). Overall ELM can well track the snow phenology, but shows a late start of snow accumulation in winter which is consistent with the underestimation of SWE and may be related to the precipitation and air temperature bias in the meteorological forcing data of

NLDAS-2 and the partitioning of precipitation into rainfall or snowfall in ELM. An earlier snowmelt is also found in ELM, and there are similar issues in other LSMs, e.g., CLM4 (Toure et al., 2018) and Noah with Multi-Parameterization (Noah-MP) (Xiao et al., 2021).



There are still some uncertainties in the benchmarking datasets used in this study. First, the MCD43 product performs well in
representing $\alpha_{sur}$ during snow cover periods, but may have poor performance for ephemeral snow due to its assumptions of
stable land surface status within 16 days (Wang et al., 2012; Wang et al., 2014). Besides, frequent cloud cover and a lack of
explicit representations of topographic effects can affect the accuracy of the MCD43 product over mountainous areas (Hao et
al., 2019; Hao et al., 2018a; Hao et al., 2018b). There are some inconsistencies between STC-MODSCAG and SPIReS (Figure
3-4), due to the different algorithms and data processing (e.g., interpolation and filtering). Although the physically-based STC-
MODSCAG and SPIReS provide higher quality unbiased $f_{sno}$ estimates than the MOD10A1 snow product based on empirical
algorithms against field measurements across different forest cover, snow cover, snow climate and viewing angles (Bair et al.,
2021c; Rittger et al., 2013; Stillinger et al., 2022), the issues of reflectance errors, one to many problems intrinsic to spectral
unmixing, cloud contamination, topographic shadows, sun-sensor geometric effects, and the impacts of forest cover can still
affect their reliabilities (Bair et al., 2021b; Raleigh et al., 2013; Stillinger et al., 2022). These issues can also affect the accuracy
of extracted snow phenology (Section 2.4). Uncertainties of $S_{sno}$ and $R_{sno}$ in STC-MODSCAG/STC-MODDRFS and SPIReS
exist (Bair et al., 2019). In summary, the heterogeneity of snow within pixels, relatively low spectral resolution, and
interference from clouds limits the diagnostic capabilities of snow properties from MODIS. Ongoing and upcoming
hyperspectral remote sensing missions (e.g., the recently launched Environmental Mapping and Analysis Program
(https://www.enmap.org/) and NASA's Surface Biology and Geology (Cawse-Nicholson et al., 2021) will enhance the abilities
of remote sensing to monitor snow properties. There are also some discrepancies between UA and SNODAS (Figure 11-12).
The uncertainties in the PRISM data over complex terrain (Henn et al., 2018) may degrade the performance of UA. Compared
to ground survey data, SWE in SNODAS over alpine areas has degraded performance due to the neglect of wind redistribution
of snow (Clow et al., 2012). Compared to GPS interferometric reflectometry snow depth data, SNODAS still needs to be
improved over complex terrain and areas with high vegetation heterogeneities (Boniface et al., 2015). The independent
comparisons also have shown the underestimations and overestimations of SNODAS (Bair et al., 2016; Dozier, 2011; Dozier
et al., 2016). Developing reliable benchmarking datasets for advancing snow modeling is still challenging but necessary
(Ménard et al., 2019).

There is significant room for improving simulations of snow processes in ELM, ranging from the input forcing data to
parameter settings and model structure. Meteorological forcing data has been demonstrated to have large impacts on snow
simulations (Günther et al., 2019). The NLDAS-2 forcing data was used to drive ELM in the study, which is rather coarse to
represent the sub-grid heterogeneity of precipitation over mountainous areas (Tesfa et al., 2020). Although NLDAS-2 has
many improvements compared to NLDAS-1 (Xia et al., 2012), there are still some spatio-temporal discontinuities in the
precipitation of NLDAS-2 (Ferguson and Mocko, 2017; Xia et al., 2019). Besides, there are still some documented systematic
precipitation and air temperature biases in NLDAS-2 especially over mountainous areas (Henn et al., 2018; O'Neill et al.,
2021; Pan et al., 2003). The $1.9° \times 2.5°$ climatological aerosol deposition data used in the ELM simulations is too coarse to





capture the fine-scale spatial variations of BC and dust, which limits the accuracy of simulating $R_{sno}$ and thus $\alpha_{sur}$. Some parameters in ELM were set empirically or from the literature, which may contain large uncertainties. For instance, in the snow cover parameterization of ELM, snow accumulation ratio and snowmelt shape factor are empirically set as fixed values without spatio-temporal variations (Swenson and Lawrence, 2012). In the SNICAR-AD snow albedo models of ELM, spherical snow grain shape, internal mixing of BC-snow and external mixing of dust-snow were set by default, which may be oversimplified (Hao et al., 2022). Further efforts are needed to calibrate these parameters using remote sensing or data assimilation products. The model structures used in different LSMs have different complexities, assumptions and simplifications (Lee et al., 2021; Magnusson et al., 2015). Some snow processes are modelled empirically, e.g., the snow cover over complex terrain was simply parameterized as a function of the standard deviation of elevation, which may explain the large biases of $f_{sno}$ (Figure 3) over mountainous areas (Swenson and Lawrence, 2012). The large uncertainty of $S_{sno}$ is relevant to the unrealistic snow aging representations in ELM (Qian et al., 2014). Some important processes are missed in ELM, such as the snow redistribution and sublimation by blowing snow (Xie et al., 2019), and the interaction between vegetation and snow, which possibly lead to the degraded performance of ELM (Section 3). Developing accurate forcing data, improving/choosing suitable snow models/parameterizations, and calibrating/optimizing model parameters are all important for accurate simulations of snow processes in LSMs.

Further studies are needed to conduct systematic diagnosis and attributions of ELM simulation biases and evaluate the ability of ELM in capturing the long-term trends and climate effects of snow. Attributing the snow simulation biases to the specific parameterizations or processes is still challenging but necessary to identify and locate the major sources of errors. Because the snow processes are coupled and impacted by each other, further sensitivity analysis and numerical experiments varying factors one at a time are needed. An international coordinated project of the intercomparison of snow schemes in Earth system models, ESM-SNOWMIP, provides a good opportunity for ELM to identify crucial processes leading to large biases in simulated snow and compare with other LSMs from local to global scales (Krinner et al., 2018; Menard et al., 2021). In this study, we found no significant increasing or decreasing trend of snow from 2001 to 2019 over the WUS for both ELM and other benchmarking datasets. However, 19 years are not long enough to characterize long-term trend of snow and analysis was not performed on discrete river basin or elevation subsets that may be experiencing change nor during the JJA time period. This study only evaluated offline land-only ELM simulations, but how ELM can capture the impacts of snow on regional climate needs further investigations by performing E3SM simulations with active land and atmosphere model.

## 5 Conclusions

Snow over the WUS plays an important role in regional climate and hydrological and ecological systems as well as human society. This study systematically evaluated the snow properties (including $\alpha_{sur}$, $\alpha_{sno}$, $f_{sno}$, $S_{sno}$, $R_{sno}$, SWE and $D_{sno}$) and snow phenology (including four snow dates and one snow duration) simulated by ELM using SNOTEL field measurements, MODIS



remote sensing products and two data assimilation products. Overall, the ELM snow simulations agree well with the
benchmarking datasets in terms of spatio-temporal distributions, interannual variabilities and elevation gradients for different
snow properties. However, ELM has large biases of $f_{sno}$ for dense forest cover and $\alpha_{sur}$ in the Rocky Mountains and Sierra
Nevada, while underestimating SWE and $D_{sno}$, especially over mountainous areas with dense forest cover for both winter and
spring. The ELM simulations shows large inconsistencies with the remote sensing retrievals of $S_{sno}$ and $R_{sno}$. Compared to
SNOTEL, ELM has larger negative biases of SWE, probably because there are some systematic biases of precipitation and air
temperature in NLDAS-2. Besides, there is a large spatial-scale mismatch between point-scale field measurements and grid-
level simulations, which can contribute to the large biases of ELM. There are also some inconsistencies of snow phenology
between ELM and remote sensing products with ELM showing later snow onset, earlier depletion and shorter snow duration,
consistent with the underestimation of SWE. This study documents the ELM performance in simulating snow processes and
demonstrates the necessity for further improving the snow properties and snow phenology represented in LSMs. Further efforts
are needed to improve the accuracy of snow properties, especially $S_{sno}$ and $R_{sno}$ in both ELM simulations and remote sensing
retrievals and resolve the early melt-off of snow in spring and underestimations of SWE in ELM especially over the complex
terrain of the WUS.

**Appendix A**

**Table A1. The acronyms and symbols used in the study.**

| Category | Abbreviation or symbol | Explanation |
|---|---|---|
| Snow property | $\alpha_{sur}$ | Surface albedo |
| | $\alpha_{sno}$ | Snow albedo |
| | $f_{sno}$ | Snow cover fraction |
| | $S_{sno}$ | Snow grain size |
| | $R_{sno}$ | Snow albedo reduction |
| | SWE | Snow water equivalent |
| | $D_{sno}$ | Snow depth |
| Snow phenology | Accumulation_onset_date | Snow accumulation onset date |
| | Depletion_onset_date | Snow cover depletion onset date |
| | Midpoint_date | Snow cover depletion midpoint date |
| | End_date | Snow end date |
| | Duration | Snow duration days |
| Model name | E3SM | Energy Exascale Earth System Model |

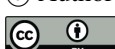



| | ELM | E3SM land model |
|---|---|---|
| | LSM | Land surface model |
| | CLASS | Canadian Land Surface Scheme |
| | CLM | Community Land Model |
| | SNICAR | the Snow, Ice, and Aerosol Radiative model |
| | SNICAR-AD | SNICAR with the delta-Eddington adding–doubling radiative transfer solver |
| | PRISM | Parameter-elevation Regressions on Independent Slopes Model |
| | Noah-MP | Noah with Multi-parameterization |
| Dataset name | MODIS | Moderate Resolution Imaging Spectroradiometer |
| | BCQC | Bias Correction and Quality Control data |
| | SNOTEL | Snow Telemetry stations |
| | STC-MODSCAG/STC-MODDRFS | the spatially and temporally complete (STC) MODIS Snow-Covered Area and Grain size/MODIS Dust and Radiative Forcing in Snow |
| | MCD43A3 | MODIS daily surface albedo version 6 product |
| | SPIReS | Snow Property Inversion From Remote Sensing product |
| | UA | University of Arizona daily snow product |
| | SNODAS | SNOw Data Assimilation System daily snow product |
| | MOD10A1 | Official MODIS snow cover product |
| | NLDAS-2 | National Land Data Assimilation System phase 2 |
| Accuracy metrics | $R^2$ | Coefficient of determination |
| | RMSE | Root mean square error |
| | IAV | Interannual variability |
| | R | Correlation coefficient |
| | rBias | Relative Bias |
| | RMSD | Root mean square deviations |
| | rRMSD | Relative RMSD |
| Others | DOY | Day of year |
| | SZA | Solar zenith angle |
| | BC | Black carbon |
| | LAP | Light-absorbing particles |



| | MK | Mann–Kendall test |
| | NASA | National Aeronautics and Space Administration |

**Table A2. Overview of some typical studies and this study on the evaluation of snow processes in land surface models (LSMs).**

| Model | Spatial resolution | Involved Snow properties | Involved snow phenology metrics | Reference |
|---|---|---|---|---|
| CLM4 | 0.5°×0.67° | $f_{sno}$, SWE, $D_{sno}$ | - | (Toure et al., 2016) |
| CLM4.5 | 0.5°×0.67° | $f_{sno}$, SWE, $D_{sno}$ | SEnDt | (Toure et al., 2018) |
| CLASS | 0.25° | $\alpha_{sur}$, $f_{sno}$, SWE | - | (Verseghy et al., 2017) |
| Noah-MP | 10 km | $\alpha_{sur}$, $f_{sno}$, $D_{sno}$ | - | (Jiang et al., 2020) |
| E3SM v1 | 1° | SWE | - | (Brunke et al., 2021) |
| ELM | 0.125° | $\alpha_{sur}$, $f_{sno}$, $S_{sno}$, $R_{sno}$, SWE, $D_{sno}$ | SOnDt, SMOnDt, SMMidDt, SEnDt, SDurDy | This study |

**Code and data availability**

ELM model codes with new updates used in the study are publicly available at https://doi.org/10.5281/zenodo.6324131. All the SRTM DEM, and GFCC forest cover, and MCD43 surface albedo data can be freely downloaded from the Google Earth Engine (GEE, https://earthengine.google.com). The STC-MODSCAG/STC-MODDRFS and SPIReS data used in the study are available at https://doi.org/10.5281/zenodo.7194702. The SPIReS code is publicly available at https://github.com/edwardbair/SPIRES. UA and SNODAS data can be accessed at https://nsidc.org/data/nsidc-0719 and https://nsidc.org/data/g02158, respectively. BCQC SNOTEL data is available at https://www.pnnl.gov/data-products. Codes to process data, generate all results and produce all figures are archived at https://github.com/daleihao/snow_evaluation_ELM.

**Author contributions**

DH conceived the study, designed the experiments, collected the evaluation data, conducted the simulations, analyzed the results, and drafted the original manuscript. GB conceived the study, designed the experiments, discussed the results, and edited the manuscript. EB, KR, and TS provided the remote sensing data and edited the manuscript. YG and LRL discussed the results and edited the manuscript. All authors contributed to reviewing and improving the manuscript.





## Competing interests

A co-author is a member of the editorial board of The Cryosphere. The peer-review process was guided by an independent editor, and the authors have no other competing interests to declare.

## Acknowledgements

This research was conducted at Pacific Northwest National Laboratory, which is operated for the U.S. Department of Energy by Battelle Memorial Institute under contract DEAC05-76RL01830. This research used resources of the National Energy Research Scientific Computing Center (NERSC), a DOE Office of Science User Facility supported by the Office of Science of the U.S. Department of Energy under contract no. DE-AC02-15 05CH11231. The reported research also used DOE's Biological and Environmental Research Earth System Modeling program's Compy computing cluster located at Pacific
Northwest National Laboratory.

## Financial support

This research has been supported by the U.S. Department of Energy, Office of Science, Office of Biological and Environmental Research, Earth System Model Development program area, as part of the Climate Process Team projects and the U.S. National Oceanic and Atmospheric Administration (NOAA, grant no. NOAA-OAR-CPO-2019-2005530). Support was also provided
by NASA grants 80NSSC20K1349, 80NSSC18K0427, NSSC22K0703, 80NSSC21K0997, and 80NSSC20K1722.




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
