# Peer review of "Evaluation of E3SM Land Model snow simulations over the Western United States"

_EGUsphere, 2022_

## Author Comment (AC1)

**Reviewer 1**

**General comments:**

The manuscript by Hao et al. (egusphere-2022-1097) evaluated the performance of E3SM land model in simulating a number of snow parameters, e.g., snow cover fraction, surface albedo, snow water equivalent and snow depth, and snow phenology using in situ, remote sensing, and reanalysis data. The paper presented a comprehensive model evaluation and thoroughly discussed the sources of the model uncertainties and biases. I think the results of the paper are useful for the related researchers to understand the capacity and drawbacks of E3SM land model in snow simulations, and to get insights of how to further improve the model.

Thank you very much for your helpful suggestions/comments!

**Specific comments:**

1.  I would suggest revising the title of the paper. As this paper mainly evaluated the performance of the E3SM land model in simulating snow processes against a number of observation datasets. Basically, this is not an evaluation of snow processes.

Agree. To avoid the misunderstanding, we have revised the title as "Evaluation of E3SM Land Model snow simulations over the Western United States" in the revised manuscript.

2.  L208-209. "The snow accumulation and snowmelt seasons are defined as the periods from September to January and from February to August, respectively." However, as I know, many regions in the Northern Hemisphere have the peak SWE in February. Please discuss more about the rationality of this definition in snow season.

In this study, we defined the snow seasons according to the typical seasonal cycles of snow cover fraction ($f_{sno}$) over the Western US (https://nsidc.org/snow-today/snow-viewer; Rittger et al., 2022) to derive the five phenology metrics. Figure R1 shows the seasonal variations of $f_{sno}$ for North America (Brutel-Vuilmet et al., 2013). Such a definition of snow seasons is consistent with previous studies (Peng et al., 2013; Anttila et al., 2018). We clarified these in Line 206-207 of the revised manuscript.

We used additional criteria to estimate/retrieve snow phenology (see Section 2.4 for details), to reduce the impacts of metric definition and data noise. Only when all the criteria were met, the corresponding pixels were used in the analysis. Under such definition of snow seasons, the phenology metrics in most pixels have been successfully retrieved (Section 3.3).

Note that this study defined snow season and phenology based on $f_{sno}$ rather than SWE. From the energy balance perspective, SWE may be better for use in defining snow phenology. Therefore,

how ELM can capture the date of peak snowpack and snowmelt timing and rate needs further investigation based on SWE. We have clarified this point in Line 481-482 of the revised manuscript.

[Figure]

Figure R1: Seasonal cycles of observed (black) and simulated (red) multi-model mean snow cover for southern (latitude < 50°N) and northern (latitude ≥ 50°N) Eurasia and North America, averaged over the 1979–2005 period (Brutel-Vuilmet et al., 2013).

3. Fig. 3. I notice that the temporal correlations between the simulated and observed snow fractions in winter are obviously lower than those in spring. Please explain the reason. Are they caused by different parameterizations of the model for the two seasons? Normally, it is more challenging for the snow models to simulate the complex snow processes during the melt season. Thus, the results in Fig. 3 are confusing. Pleas add more discussions.

ELM uses the same snow cover parameterizations for different seasons, where the hysteresis of snow accumulation and ablation is accounted for in ELM (Swenson and Lawrence, 2012). Indeed, for temporal correlation, ELM has a low correlation in the mountainous areas with remote sensing data in winter, but has a relatively high correlation with them in spring This is possibly caused by the smaller change of snow cover fraction in winter than spring (Figure R1). We have clarified this in Line 275-276 of the revised manuscript.

Although spring has a higher temporal correlation than winter, the statistics show that ELM has a higher relative accuracy in winter than spring. For instance, the relative RMSD values are 39.5 and 39.1 %, respectively compared to STC-MODSCAG and SPIReS in winter, while those

values are 57.8 and 46.6 %, respectively in spring. Our analysis of the snow phenology also demonstrated that accurately simulating the snowmelt processes is challenging in ELM. We have clarified these in Line 274 and 439-443 of the revised manuscript.

4. Table 2. Some simulated snow variables showed small correlations with some observations (R=-0.2~0.2) but obviously higher correlations with other observations (R>0.5). Please explain the reason.

Indeed, for snow cover fraction, surface albedo, SWE, and snow depth, ELM simulations have a high spatial correlation with the observations/assimilated data. All of these variables show significant spatial patterns and elevation gradients with large values in the high snow-covered regions and small values in the low snow-covered regions. Such spatial patterns are easily captured by ELM. We have discussed these in Line 466-468 of the revised manuscript. Besides, the models/parameterizations in ELM related to these variables have been calibrated or well validated based on remote sensing and field data (Swenson and Lawrence, 2012). We have introduced these in Section 2.1 of the revised manuscript.

By contrast, for snow grain size, snow albedo reduction and snow albedo, there are still low spatial correlations between ELM simulations and benchmarking datasets, which are related to the uncertainties in the model simulations and remote sensing datasets:

For the model simulations: 1) The 1.9°×2.5° climatological aerosol deposition data used in the ELM simulations is too coarse to capture the fine-scale spatial variations of BC and dust, which limits the accuracy of simulated $R_{sno}$. 2) In the SNICAR-AD snow albedo models of ELM, spherical snow grain shape, internal mixing of BC-snow and external mixing of dust-snow were set by default, which can potentially affect the accuracy of $R_{sno}$. 3) The large uncertainty of $S_{sno}$ is relevant to the unrealistic snow aging representations in ELM (Qian et al., 2014). All of the above-mentioned uncertainties can further affect simulated $\alpha_{sno}$. We have discussed these in Line 509-534 of the revised manuscript.

For the remote sensing data: The heterogeneity of snow within pixels, relatively low spectral resolution, and interference from clouds limit the accuracy of snow properties diagnosed from MODIS. We have discussed these in Line 484-497 of the revised manuscript.

**Technical corrections:**

1. L264-265. "The regional average $f_{sno}$ is 0.41 and 0.15, respectively for spring and winter." However, Fig. 3 shows winter has higher $f_{sno}$ than spring. Please recheck whether it is a typo.

Sorry for the typo. It has been corrected.

2. Fig. 3. I would prefer using blue for areas having more snow and using red for snow-rare areas in figures.

Considering that we used the same color schemes for all the spatial maps of different snow properties and phenology metrics, we'd prefer not to change the color scheme for this particular figure in order to keep the color schemes consistent throughout the manuscript.

3. Figure captions. I would suggest giving the full names of the variables in the captions, instead of abbreviations.

Good suggestion! We have modified the corresponding captions in the figures and tables throughout the manuscript.

.

4. Table 2. It is likely the typesetting of Table 2 is problematic. It is not easy to match the products with the corresponding error metrics. Please improve.

The table has been revised.

**References**

Anttila, K., Manninen, T., Jääskeläinen, E., Riihelä, A., and Lahtinen, P.: The Role of Climate and Land Use in the Changes in Surface Albedo Prior to Snow Melt and the Timing of Melt Season of Seasonal Snow in Northern Land Areas of 40°N–80°N during 1982–2015, Remote Sensing, 10, 1619, 2018.

Brutel-Vuilmet, C., Ménégoz, M., and Krinner, G.: An analysis of present and future seasonal Northern Hemisphere land snow cover simulated by CMIP5 coupled climate models, The Cryosphere, 7, 67–80, https://doi.org/10.5194/tc-7-67-2013, 2013.

Peng, S., Piao, S., Ciais, P., Friedlingstein, P., Zhou, L., and Wang, T.: Change in snow phenology and its potential feedback to temperature in the Northern Hemisphere over the last three decades, Environmental Research Letters, 8, 014008, 2013.

Qian, Y., Wang, H., Zhang, R., Flanner, M. G., and Rasch, P. J.: A sensitivity study on modeling black carbon in snow and its radiative forcing over the Arctic and Northern China, Environmental Research Letters, 9, 064001, 2014.

Rittger, K., Brodzik, M.J., & Raleigh, M. Snow Today. Boulder, Colorado USA. National Snow and Ice Data Center, 2022.

Swenson, S. C. and Lawrence, D. M.: A new fractional snow-covered area parameterization for the Community Land Model and its effect on the surface energy balance, Journal of Geophysical Research: Atmospheres, 117, 2012.

---

## Author Comment (AC2)

**Reviewer 2**

This manuscript evaluated the performance of ELM in simulating snow-related properties across the Western United States. The authors conducted a 50-year offline regional land simulation and evaluated the modeled snow properties against various observational and reanalysis datasets. The experiment is well-designed, and the discussions are well-presented. Such comprehensive model evaluations are valuable for further studies on improving climate simulations, especially for studies based on E3SM.

Thank you very much for your helpful suggestions/comments!

General comments:

1. The authors should consider revising the title of this manuscript. Snow processes on land imply snow metamorphism, i.e., how snow changes over time. This manuscript focuses more on the accuracy of ELM-simulated snow properties rather than evaluating the ELM snow metamorphism schemes.

Considering this comment and the 1st comment from Reviewer 1, to avoid the misunderstanding, we have revised the title as "Evaluation of E3SM Land Model snow simulations over the Western United States" in the revised manuscript.

2. The authors gathered a lot of observational data to evaluate ELM model simulations. They presented many well-organized figures, including model-minus-observation and their temporal correlations for each snow property. Yet, they need to add more discussions on how these properties interact. For example, is the bias in snow grain size contributing to snow albedo and further influencing SWE and snow-covered fraction? Such discussions are crucial and will help the users to understand the snow simulations in ELM.

Good point! Systematic analysis of the propagation of uncertainty of different snow properties is crucial for better understanding the uncertainty source of snow simulations in ELM. Both snow grain size ($S_{sno}$) and snow albedo reduction ($R_{sno}$) are related to snow albedo ($\alpha_{sno}$). $\alpha_{sur}$ is a combination of $\alpha_{sno}$ and the albedo of non-snow vegetation/soil. $\alpha_{sno}$ determines the energy absorbed by snow and thus affects the snowmelt process. Therefore, SWE and snow depth ($D_{sno}$) can be further affected by $\alpha_{sno}$ simulations. In ELM, snow cover fraction ($f_{sno}$) is modeled as a function of SWE. We have added such discussion on connecting different snow properties and the corresponding uncertainties as below in Line 519-530 of the revised manuscript.

The climatological aerosol deposition data used in the ELM simulations is too coarse to capture the fine-scale spatial variations of BC and dust, which limits the accuracy of simulated $R_{sno}$ and thus $\alpha_{sno}$. The model structures used in different LSMs have different complexities, assumptions and simplifications. In ELM, some snow processes are modelled empirically, and some

parameters are set empirically or from the literature, which may contain large uncertainties. For instance, in the ELM snow albedo model, spherical snow grain shape, internal mixing of BC-snow and external mixing of dust-snow are the default settings, which can potentially affect the accuracy of $R_{sno}$ and $\alpha_{sno}$. The large uncertainty of $S_{sno}$ is a factor responsible for the unrealistic snow aging representations in ELM (Qian et al., 2014), which can further affect $\alpha_{sno}$. The bias in $\alpha_{sno}$ can further affect the accuracy of absorbed energy by snow and $\alpha_{sur}$ (contains the contributions from snow and non-snow vegetation/soil), and thus the change of SWE and $D_{sno}$. The uncertainties of SWE can further propagate to $f_{sno}$, because ELM uses SWE to estimate $f_{sno}$ (Swenson and Lawrence, 2012). In the snow cover parameterization of ELM, snow accumulation ratio and snowmelt shape factor are empirically set as fixed values without spatio-temporal variations (Swenson and Lawrence, 2012), which can also affect the accuracy of $f_{sno}$. The snow cover over complex terrain is simply parameterized as a function of the standard deviation of elevation, which may explain the large biases of $f_{sno}$ (Figure 3) over mountainous areas (Swenson and Lawrence, 2012). All of these contribute to the bias of snow phenology in ELM (Section 3.2).

3. Lastly, the authors conducted an offline ELM experiment, presumably for computational efficiency. Would the results differ with coupled simulations considering various snow-related feedbacks?

Yes, coupled simulations can consider the snow-related feedback between land and atmosphere, but can also have more uncertainties from the atmospheric forcing which is simulated. We conducted the offline land-only simulations for the following reasons:

1) Errors in both simulated temperature and precipitation have been recognized as the main drivers of snowpack errors in the coupled E3SM simulations (Brunke et al., 2021). To reduce the impacts of the uncertainties from atmospheric forcing, our study focused on evaluating the offline ELM simulations forced by data from the National Land Data Assimilation System phase 2 (NLDAS-2). However, snow-related land-atmosphere interactions were neglected in these land-only simulations. How well E3SM capture the impacts of snow on regional climate needs further investigations by performing coupled E3SM simulations with active land and atmosphere models. We added these discussions in Line 545-550 of the revised manuscript.

2) We conducted the ELM simulations over the WUS at 0.125° spatial resolution, which is higher than the typical 1° spatial resolution used in fully coupled E3SM simulations (Golaz et al., 2022). We expect that such high-resolution simulations can better capture the spatial heterogeneity of snowpack. Coupled simulations at such high resolution are more computationally demanding. We have added these discussions in Section 2.2 of the revised manuscript.

**References**

Brunke, M. A., Welty, J., and Zeng, X.: Attribution of Snowpack Errors to Simulated Temperature and Precipitation in E3SMv1 Over the Contiguous United States, Journal of Advances in Modeling Earth Systems, 13, e2021MS002640, 2021.

Golaz, J.-C., Van Roekel, L. P., Zheng, X., Roberts, A. F., Wolfe, J. D., Lin, W., et al.: The DOE E3SM Model version 2: Overview of the physical model and initial model evaluation, Journal of Advances in Modeling Earth Systems, 14, e2022MS003156, 2022. https://doi.org/10.1029/2022MS003156

Qian, Y., Wang, H., Zhang, R., Flanner, M. G., and Rasch, P. J.: A sensitivity study on modeling black carbon in snow and its radiative forcing over the Arctic and Northern China, Environmental Research Letters, 9, 064001, 2014.

Swenson, S. C. and Lawrence, D. M.: A new fractional snow-covered area parameterization for the Community Land Model and its effect on the surface energy balance, Journal of Geophysical Research: Atmospheres, 117, 2012.

---

## Author Comment (AC4)

Dear Editors and Reviewers,

Thank you very much for your detailed and constructive comments and suggestions for our manuscript. We found the comments very helpful for improving our manuscript and have revised the manuscript accordingly.

Below are the major revisions that we have made:

1. We have added more discussion about the snow season definitions, model evaluation on the spatio-temporal correlation, propagation of uncertainty in the snow variables, and the potential differences between land-only and coupled simulations.
2. We have revised the title as "**Evaluation of E3SM Land Model snow simulations over the Western United States**" according to the comments from Reviewers 1 and 2.

The itemized responses to the comments and suggestions from the editors and reviewers are provided below in blue font. We hope our responses address the concerns raised in the last round of review and look forward to your decision on the publication of our manuscript.

Best regards,

Dalei Hao (on the behalf of all authors)

---

## Editor Decision (ED1)

2023-01-16

Submission egusphere-2022-1097

Thank you for your submission to be considered for publication in The Cryosphere. The authors addressed all main points in their review, which now includes a thorough evaluation of the E3SM land model. Some of the main criticism underlined the lack of discussion on peak SWE temporal variability at this scale and the linkages between snow geophysical properties and their impact on SWE and snow cover fraction.

In their response, the authors did include additional discussions that really improve the paper.

I therefore feel all main concerns have bee addressed and the paper can be published.

Regards,

Prof. Dr. Alexandre Langlois

Associate editor, *The Cryosphere*